# Testing the intrinsic mechanisms driving the dynamics of Ross River Virus across Australia

**Iain S. Koolhof**[1,2]*, **Nicholas Beeton**[3], **Silvana Bettiol**[1], **Michael Charleston**[2], **Simon M. Firestone**[4], **Katherine Gibney**[5], **Peter Neville**[6], **Andrew Jardine**[6], **Peter Markey**[7], **Nina Kurucz**[7], **Allan Warchot**[7], **Vicki Krause**[7], **Michael Onn**[8], **Stacey Rowe**[5], **Lucinda Franklin**[5], **Stephen Fricker**[9], **Craig Williams**[9], **Scott Carver**[2,10,11]

1 College of Health and Medicine, Tasmanian School of Medicine, University of Tasmania, Hobart, Tasmania, Australia, 2 College of Sciences and Engineering, School of Natural Sciences, University of Tasmania, Hobart, Tasmania, Australia, 3 Data61, CSIRO, Hobart, Tasmania, Australia, 4 Melbourne Veterinary School, Faculty of Science, University of Melbourne, Melbourne, Victoria, Australia, 5 Victorian Department of Health and Human Services, Communicable Disease Epidemiology and Surveillance, Health Protection Branch, Melbourne, Victoria, Australia, 6 Department of Health, Western Australia, Environmental Health Directorate, Public and Aboriginal Health Division, Perth, Western Australia, Australia, 7 Centre for Disease Control, Northern Territory Department of Health, Northern Territory, Darwin, Australia, 8 Brisbane City Council, Brisbane, Queensland, Australia, 9 Australian Centre for Precision Health, University of South Australia, Adelaide, South Australia, Australia, 10 Odum School of Ecology, University of Georgia, Georgia, United States of America, 11 Center for the Ecology of Infectious Diseases, University of Georgia, Georgia, United States of America

* koolhofi@utas.edu.au

**Data Availability Statement:** The data underlying the results presented in the study are available from the following sources: RRV Human

## Abstract

The mechanisms driving dynamics of many epidemiologically important mosquito-borne pathogens are complex, involving combinations of vector and host factors (e.g., species composition and life-history traits), and factors associated with transmission and reporting. Understanding which intrinsic mechanisms contribute most to observed disease dynamics is important, yet often poorly understood. Ross River virus (RRV) is Australia's most important mosquito-borne disease, with variable transmission dynamics across geographic regions. We used deterministic ordinary differential equation models to test mechanisms driving RRV dynamics across major epidemic centers in Brisbane, Darwin, Mandurah, Mildura, Gippsland, Renmark, Murray Bridge, and Coorong. We considered models with up to two vector species (*Aedes vigilax*, *Culex annulirostris*, *Aedes camptorhynchus*, *Culex globocoxitus*), two reservoir hosts (macropods, possums), seasonal transmission effects, and transmission parameters. We fit models against long-term RRV surveillance data (1991–2017) and used Akaike Information Criterion to select important mechanisms. The combination of two vector species, two reservoir hosts, and seasonal transmission effects explained RRV dynamics best across sites. Estimated vector-human transmission rate (average $\beta = 8.04 \times 10^{-4}$ per vector per day) was similar despite different dynamics. Models estimate 43% underreporting of RRV infections. Findings enhance understanding of RRV transmission mechanisms, provide disease parameter estimates which can be used to guide future research into public health improvements and offer a basis to evaluate mitigation practices.

Notification Data: The Department of Health and the Western Australian Notifiable Infectious Diseases Database (https://www.health.wa.gov.au/~/media/Files/Corporate/general%20documents/Performance/Information%20access/Information-Release-Form.xls); the Northern Territory Department of Health and Centre of Disease Control Surveillance Unit (https://health.nt.gov.au/data-and-research/health-data/health-data?SQ_PAINT_LAYOUT=664820&SQ_ASSET_CONTENTS=&curr=578946&print=yes); the Queensland Department of Health and the Communicable Disease Branch (https://www.health.qld.gov.au/clinical-practice/guidelines-procedures/diseases-infection/surveillance/reports/notifiable/data-request); the Victorian Department of Health and the Communicable Disease Epidemiology and Surveillance Branch (https://www.health.vic.gov.au/infectious-diseases/about-infectious-diseases-surveillance-in-victoria); the Department of Health South Australia and the Communicable Disease Control Branch (https://www.sahealth.sa.gov.au/wps/wcm/connect/public+content/sa+health+internet/about+us/department+for+health+and+wellbeing/health+regulation+and+protection/communicable+disease+control+branch/communicable+disease+control+branch). Mosquito Surveillance Data: The Arbovirus Surveillance and Research Laboratory for Western Australia (https://www.health.wa.gov.au/articles/a_e/arbovirus-surveillance-program), The Centre for Disease Control for the Northern Territory (https://health.nt.gov.au/professionals/centre-for-disease-control/cdc-contacts); the Mosquito Management and Asset Services in the Brisbane City Council for Queensland (https://www.brisbane.qld.gov.au/clean-and-green/natural-environment-and-water/biodiversity-in-brisbane/wildlife-in-brisbane/invasive-plants-and-animals/mosquitoes); the Department of Primary Industries for Victoria (https://www.betterhealth.vic.gov.au/health/healthyliving/mosquito-management-victoria); and the University of South Australia for South Australia (https://www.sahealth.sa.gov.au/wps/wcm/connect/public+content/sa+health+internet/public+health/pest+management/mosquitoes/arbovirus+and+mosquito+monitoring+reports & Email: mozzie.monitors@unisa.edu.au). Human Population Data: Australian Bureau of Statistics (https://dbr.abs.gov.au/).

**Funding:** Financial support for this project was provided through the Funding Initiative for Mosquito Management In Western Australia Research Grants (to SC and ISK), administered by the Department of Health, Western Australia. The funders had no role in study design, data collection

## Author summary

Ross River virus (RRV) causes the highest number of vector-borne disease infections in Australia, yet the mechanisms driving its transmission across regions remains poorly understood. We analyzed long-term surveillance data from eight epidemic regions spanning tropical to temperate climates. We tested the importance of different mosquito vectors, wildlife hosts, and varies seasonal and transmission effects in explaining observed patterns of RRV notifications. Despite differing environments, models indicate combinations of two key mosquito vectors and two marsupial hosts, and interacting seasonally best explains RRV dynamics. Estimated mosquito-human transmission rates were similar across regions, whereas wildlife contributions varied. Models estimate 43% underreporting of RRV infections nationally. Findings provide new quantitative insights on transmission mechanisms and health impacts of RRV. Estimating mechanisms and key parameters allows for the future assessment of public health interventions like mosquito control. This modelling framework evaluating long-term data could be applied to other complex vector-borne diseases to unravel intrinsic drivers and guide mitigation strategies.

## Introduction

Spillover into humans from non-human animal sources is a common characteristic of many vector-borne pathogens, including those classified as emerging or resurging. Despite the significant public health burden of vector-borne pathogens, the specific intrinsic mechanisms driving spillover and outbreak events (beyond increases in vector abundance) are however often poorly understood. Fundamentally, this is because the epidemiological dynamics of vector-borne diseases are complex, including a web of extrinsic (environmental and meteorological) and intrinsic (vector, host, and pathogen biology) factors, as well as features from public health (e.g., vector and host control, diagnostic standards, public behaviour, and education). This complexity presents a significant challenge to understanding the combinations of mechanisms most responsible for observed disease dynamics, yet a deeper understanding of the intrinsic mechanisms underscoring human incidence of vector-borne zoonoses is important to guide and improve public health management.

Ross River virus (RRV; family: *Togaviridae*, genus: *Alphavirus*) has the highest incidence of any vector-borne disease in Australia and is a classic illustration of complex mechanisms driving diverse disease dynamics [1–4]. The annual notification incidence rate for RRV is > 40 per 100,000 population, with an estimated annual health care and lost productivity cost of $15 million [1,5]. There are 42 known mosquito species (spanning seven genera) capable of transmitting RRV, with the primary vector mosquito species responsible for human transmission being *Aedes vigilax*, *Culex annulirostris*, *Aedes camptorhynchus*, and *Aedes notoscriptus* [6]. Several key reservoir hosts contribute to RRV transmission, all of which vary in relative competence and abundance across epidemic regions [4,7,8]. Marsupials (e.g., kangaroos, wallabies, & possums) are generally considered the most competent and important host reservoirs of RRV. However, placental mammals and birds may also contribute to transmission, adding to the ecological complexity in mechanisms driving disease occurrence [4]. RRV is sustained enzootically between mosquitoes and non-human mammals, and evidence indicates that epizootics lead to spillover and human epidemics [3,9–11]. Transmission dynamics of RRV are also variable across geographic regions, often linked with the effects of climate on mosquito species compositions and abundance [2,8,12,13]. Changes in climate can drive shifts in species community compositions and the dominance of specific species. Clinical symptoms of RRV in

and analysis, decision to publish, or preparation of the manuscript.

**Competing interests:** The authors have declared that no competing interests exist.

humans include fever, polyarthralgia, rashes, polyarthritis, lymphadenopathy, lethargy, headaches, myalgias, photophobia, and glomerulonephritis [1,5,6]. The duration and extent to which symptoms persist vary between individuals, however, a typical period of morbidity extends from 3–6 months but, in some instances, can exceed one year [5,6].

This study combines diverse and extensive data to advance understanding of the intrinsic mechanisms that underpin mosquito-borne viral transmission associated with spillover and human incidence. We focus on RRV, owing to the exceptionally high-quality national surveillance of human notifications and mosquito vector surveillance spanning multiple decades, across multiple epidemic centres in Australia. This rich epidemiological surveillance of long-term human RRV incidence has been crucial in establishing valuable predictive disease surveillance systems for outbreak detections, particularly using environmental and vector populations to aid in public health management. However, appreciation of which intrinsic factors are most important in epidemic centres has remained elusive (e.g., the most important vector species, host species and other mechanisms governing human incidence at a site and across sites). Furthermore, advancing mechanistic understanding of human RRV incidence is essential because climatic effects and vector abundance do not always predict human outbreaks [1,13–16].

Beyond vector and host species abundance, a range of other intrinsic mechanisms can also impact the dynamics of RRV. For example, waning host immunity may be key for epizootics, spillover, and epidemics to occur [7]. This may explain why vector abundance is not always a reliable indicator of epidemics [17,18]. Seasonal variation in vector feeding rates (associated with temperature) or host preferences also has the potential to influence seasonality in transmission. Vertical transmission in vectors can also initiate and sustain local RRV transmission. Reservoir hosts may also intermittently shed the virus if the virus is able to occasionally escape immune control (recrudescence), such as when the host is under stressful conditions that result in immune suppression [19]. The relative importance of these potential factors may also vary geographically, such as among tropical northern and temperate southern epidemic centres in Australia. Collectively, there is a need for a deeper understanding of the relative importance of varying intrinsic mechanisms that drive observed dynamics of RRV in Australia–a problem common to other vector-borne zoonoses of public health importance globally. By combining the long-term surveillance of human incidence of vector-borne disease and vector populations with information on reservoir host communities and other intrinsic factors, research can begin to test potential mechanisms driving variation in the pattern of human incidence and contrast these epidemiological systems across epidemic centres.

In this study, we bring together, to the best of our knowledge, the most extensive and diverse data on a vector-borne disease in Australia to assess the relative potential of hypothesised transmission mechanisms influencing the dynamics of RRV incidence. We utilise long-term empirical data on human and vector populations and integrate these, with other vector/host/pathogen parameters, into multi-vector, multi-host, multi-transmission mechanistic Susceptible-Infected-Recovered (SIR) models. We fit our models against high-quality long-term RRV surveillance data from humans, estimate a range of critical parameters (e.g., transmission rates), and use model selection to evaluate the importance of varying mechanisms involved in disease transmission. This study focuses on estimating parameters associated with RRV transmission dynamics and evaluating model fits to observed data retrospectively. Our aim is not to develop a prospective forecasting tool per se, but rather to use the extensive long-term data to gain ecological insights into transmission mechanisms. While forecast modelling approaches are important for prediction, here we take advantage of the rich dataset to look retrospectively and build mechanistic understanding of the drivers and parameters underlying observed dynamics. The primary aims of this study are to (1) evaluate the relative likelihood of hypothesised mechanisms responsible for observed human RRV dynamics; (2) estimate key

parameters associated with the transmission of RRV and reporting of RRV notifications; and (3) assess how hypothesised mechanisms driving RRV dynamics vary, or are similar, among epidemic sites around Australia.

## Methods

### Model structure

In this study, we use deterministic Susceptible-Infected-Recovered ordinary differential equation (ODE) models [20] to investigate the ecological mechanisms governing RRV for each study site. There were eleven ecological scenarios fitted to the RRV notification data using a variety of vector, host, and transmission parameters to investigate mechanisms driving RRV transmission (Please refer to the Model selection section below). Our full model includes up to two vector species (see Vector Monitoring and Competence Data below), three host species (see Reservoir Host Population and Competence Data below), and multiple mechanisms to potentially shape transmission (System of Eq 1). Accordingly, the transmission of RRV in our model consists of up to five populations: three host species and two mosquito species, with mosquito vectors divided into two subgroups of susceptible ($S$), infected ($I$), and hosts divided into three subgroups susceptible, infected, and recovered ($R$) classes. The rate of RRV transmission from mosquitoes to hosts is assumed to be frequency-dependent, whereby the rate of transmission increases with the total proportion of the vector population which is infected and therefore allowing for disease transmission to persist even with low host densities [7,21]. The rate of RRV transmission from hosts to mosquitoes is assumed to be density-dependent, whereby the contact rate between susceptible and infected mosquitoes with hosts depends upon both population densities, with higher densities increasing transmission rates [22].

The full model is described mathematically as follows

$$N_{v_1} = S_{v_1} + I_{v_1}$$

$$N_{v_2} = S_{v_2} + I_{v_2}$$

$$N_k = S_k + I_k + R_k$$

$$N_p = S_p + I_p + R_p$$

$$N_h = S_h + I_h + R_h$$

$$\frac{dS_{v_1}}{dt} = b(t)_{v_1}(1 - \phi_{v_1})N_{v_1} - \beta_\vartheta \left( \beta_{kv_1}\left(\frac{S_{v_1}I_k}{N_{v_1}}\right) - \beta_{pv_1}\left(\frac{S_{v_1}I_p}{N_{v_1}}\right) - \beta_{hv_1}\left(\frac{S_{v_1}I_h}{N_{v_1}}\right) \right) - d_{v_1}S_{v_1}$$

$$\frac{dI_{v_1}}{dt} = b(t)_{v_1}\phi_{v_1}N_{v_1} + \beta_\vartheta \left( \beta_{kv_1}\left(\frac{S_{v_1}I_k}{N_{v_1}}\right) + \beta_{pv_1}\left(\frac{S_{v_1}I_p}{N_{v_1}}\right) + \beta_{hv_1}\left(\frac{S_{v_1}I_h}{N_{v_1}}\right) \right) - d_{v_1}I_{v_1}$$

$$\frac{dS_{v_2}}{dt} = b(t)_{v_2}(1 - \phi_{v_2})N_{v_2} - \beta_\vartheta \left( \beta_{kv_2}\left(\frac{S_{v_2}I_k}{N_{v_2}}\right) - \beta_{pv_2}\left(\frac{S_{v_2}I_p}{N_{v_2}}\right) - \beta_{hv_2}\left(\frac{S_{v_2}I_h}{N_{v_2}}\right) \right) - d_{v_2}S_{v_2}$$

$$\frac{dI_{v_2}}{dt} = b(t)_{v_2}\phi_{v_2}N_{v_2} + \beta_\vartheta\left(\beta_{kv_2}\left(\frac{S_{v_2}I_k}{N_{v_2}}\right) + \beta_{pv_2}\left(\frac{S_{v_2}I_p}{N_{v_2}}\right) + \beta_{hv_2}\left(\frac{S_{v_2}I_h}{N_{v_2}}\right)\right) - d_{v_2}I_{v_2}$$

$$\frac{dS_k}{dt} = b(t)_{\vartheta k}\left(N_k - d_kS_k - \beta_{v_1k}S_kI_{v_1} - \beta_{v_2k}S_kI_{v_2}\right)$$

$$\frac{dI_k}{dt} = \rho(t)_{\vartheta k}\left(S_k\left(\beta_{v_1k}\frac{I_{v_1}}{N_{v_1}} + \beta_{v_2k}\frac{I_{v_2}}{N_{v_2}}\right)\right) - (\gamma_k + d_k)I_k + \omega_kR_k$$

$$\frac{dR_k}{dt} = \gamma_kI_k - (d_k + \omega_k)R_k$$

$$\frac{dS_p}{dt} = b(t)_{\vartheta p}\left(N_p - d_pS_p - \beta_{vp}S_pI_{v_1} - \beta_{v_2p}S_pI_{v_2}\right)$$

$$\frac{dI_p}{dt} = \rho(t)_{\vartheta p}\left(S_p\left(\beta_{v_1p}\frac{I_{v_1}}{N_{v_1}} + \beta_{v_2p}\frac{I_{v_2}}{N_{v_2}}\right)\right) - \left(\gamma_p + d_p\right)I_p + \omega_pR_p$$

$$\frac{dR_p}{dt} = \gamma_pI_p - \left(d_p + \omega_p\right)R_p$$

$$\frac{dS_h}{dt} = b(t)_hN_h - d_hS_h - \beta_{v_1h}S_hI_{v_1} - \beta_{v_2h}S_hI_{v_2}$$

$$\frac{dI_h}{dt} = \rho(t)_{\vartheta h}\left(S_h\left(\beta_{v_1h}\frac{I_{v_1}}{N_{v_1}} + \beta_{v_2h}\frac{I_{v_2}}{N_{v_2}}\right)\right) - (\gamma_h + d_h)I_h$$

$$\frac{dR_h}{dt} = \gamma_hI_h - d_hR_h \tag{1}$$

In the above, $t$ is time, $S_i$ is the number of animals within species $i$ that are susceptible; $I_i$ the number that are infected; $R_i$ the number that are recovered. Host and vector species $i$ are defined as follows: $k$ for kangaroos/wallabies; $p$ for possums; $h$ for humans; $v_1$ for primary vector; and $v_2$ for secondary vector, respectively. The seasonally varying birth rate of species $i$ is represented by $b(t)_i$; $\phi_i$ the vertical transmission rate; $d_i$ the mortality rate; $\beta_{vi}$ the transmission rate from vector $v_1$ to host species $i$; $\rho(t)_{\vartheta i}$ the seasonally varying disease transmission rate to host species $i$; $\gamma_i$ the recovery rate of species $i$; $N_i$ the population size; and $\omega_i$ the number of species $i$ that are recrudescent.

Our transmission models consider seasonal phases and amplification effects of vector feeding on RRV transmission to host species $i$ expressed in Eq 2, denoted by $\rho(t)_{\vartheta i}$. In Eq 2, two seasonal parameters are estimated, the phase, denoted by $\epsilon_{\mu i}$, and amplitude of seasonality, denoted by $\sigma_{\varphi i}$, of RRV transmission and used in the transmission model in Eq 1. For regions and species where seasonal variation was not relevant (see Table 1), $\sigma_{\varphi i}$ was fixed to zero.

$$\rho(t)_{\vartheta i} = 1 + \sigma_{\varphi i}sin(2\pi(t - \epsilon_{\mu i})) \tag{2}$$

**Table 1. Site-specific invariant parameters of host and vector species, seasonal birth rates (range), and site characteristics.**

| Parameter | Definition | Darwin (NT) | Brisbane (Qld) | Mandurah (WA) | Mildura (Vic) | Gippsland (Vic) | Renmark (SA) | Murray Bridge (SA) | Coorong (SA) |
|---|---|---|---|---|---|---|---|---|---|
| $N_k$ | Macropod density (per km$^2$) | 8 | 4.7 | 21.9 | 1.91 | 10.7 | 4.52 | 4.52 | 4.52 |
| $N_p$ | Possum density (per km$^2$) | 300 | 34.7 | 5.7 | 100.6 | 19.6 | 11.1 | 12.1 | 1.0 |
| $\theta_k$ | Phase between macropod births (proportion of years) | 0.25 (0.22–0.29) | 0.48 (0.42–0.51) | 0.37 (0.32–0.41) | 0.54 (0.51–0.58) | 0.30 (0.27–0.33) | 0.54 (0.51–0.57) | 0.23 (0.20–0.26) | 0.09 (0.06–0.12) |
| $\theta_p$ | Phase between possum births (proportion of years) | 0.81 (0.62–1.00) | 0.13 (0.05–0.20) | 0.51 (0.42–0.58) | 0.43 (0.37–0.49) | 0.31 (0.25–0.37) | 0.43 (0.37–0.49) | 0.24 (0.18–0.30) | 0.11 (0.05–0.17) |
| $\delta_k$ | Seasonal reproduction | No | Yes | Yes | Yes | Yes | Yes | Yes | Yes |
| $\delta_p$ | Seasonal reproduction | No | Yes | Yes | Yes | Yes | Yes | Yes | Yes |
| $k$ | Macropod species | AW | EGK | WGK | EGK | EGK | RK | RK | RK |
| $p$ | Possum species | BTP | BTP | BTP | BTP | BTP | BTP | BTP | BTP |
| $v_1$ | Primary vector species | AeV | AeV | AeC | CuA | AeC | AeC | AeC | AeC |
| $v_2$ | Secondary vector species | CuA | CuA | AeV | AeC | CuG | CuA | CuA | CuG |
| *Site characteristics* | Area size (km$^2$) | 92 | 1343 | 2948 | 512 | 2229 | 916 | 1832 | 8833 |
| | Number of households | - | 16661 | - | 18393 | 15409 | 3627 | 7935 | 2107 |

It is noted that human epidemics of RRV are not always preceded by high vector abundance and that these epidemics may be driven by a decline in seroprevalence within host populations and the recruitment of new non-immune hosts, amplifying transmission [11,23–25]. Seasonally forced birth rates were used to account for seasonal vector feeding and the timing of recruitment of susceptible newborn hosts, seen in Eq 3. The relative phase for species *i* was estimated, denoted by $\theta_i$, by using nonlinear least-squares to fit the proportion of the host population that breed per month for each host and lagged to adjust for the time it takes for new offspring to leave the pouch and become accessible to feeding vectors. This was modelled by a sinusoidal function of the proportion of months within a year (Eq 3). Phase between host breeding is represented as a proportion in years, where susceptible host offspring are exposed to feeding vectors. Additionally, vector and host populations are assumed to be stable and unlikely to become extinct over time.

$$b(t)_{v_1} = (1 - \delta_{v_1} \sin(2\pi t)) d_{v_1}$$

$$b(t)_{v_2} = (1 - \delta_{v_2} \sin(2\pi t)) d_{v_2}$$

$$b(t)_k = (1 - \delta_k \sin(2\pi(t - \theta_k))) d_k \qquad (3)$$

$$b(t)_p = (1 - \delta_p \sin(2\pi(t - \theta_p))) d_p$$

$$b(t)_h = d_h$$

## Study sites

RRV transmission was modelled across eight sites from five States and Territories in Australia, selected based on their high mosquito-borne disease attack rates, mosquito surveillance programs, spatial and epidemiological differences, and epidemiological importance [1,6–8,17,23,26,27]. The eight study sites span tropical, subtropical, and temperate climates and are likely to have different mosquito species compositions and dynamics driving local RRV transmission. Sites include Mandurah (Western Australia), Darwin (Northern Territory), Brisbane

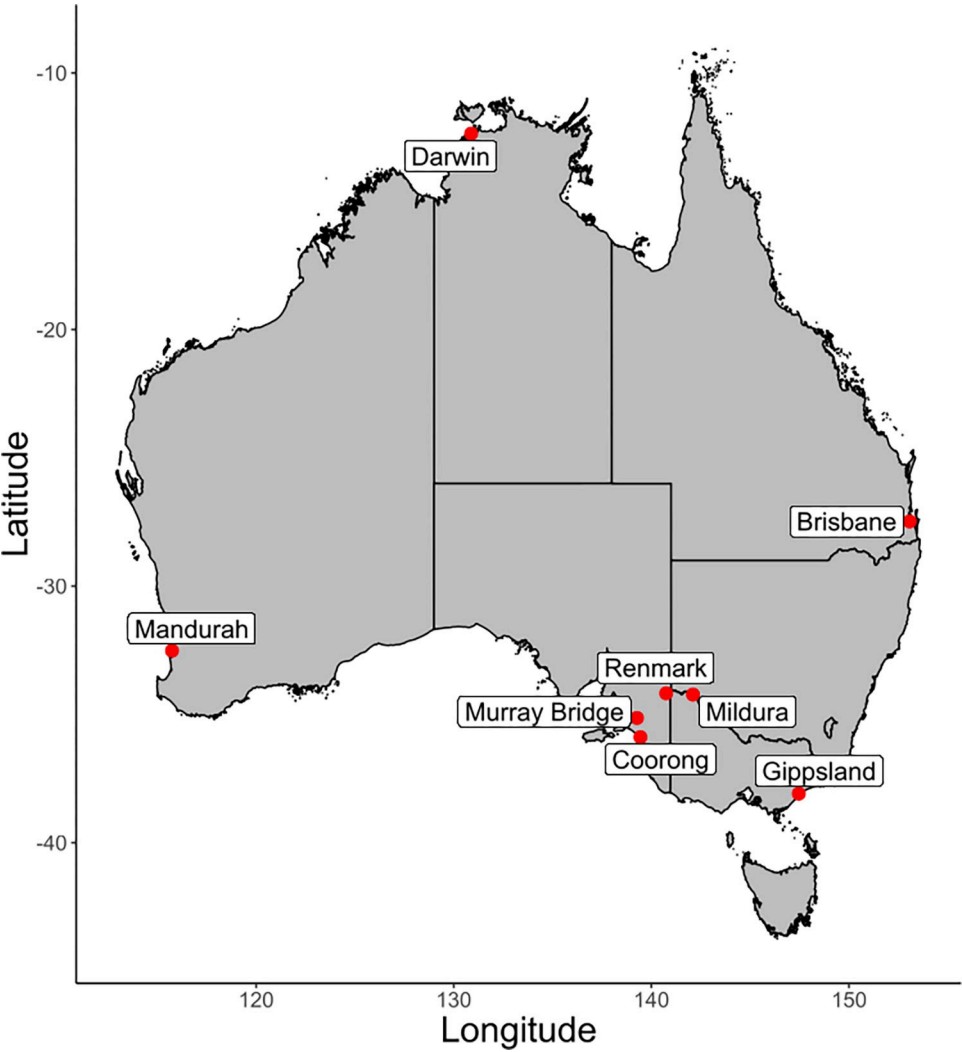

**Fig 1. Map location of study sites, showing Darwin from a tropical region, Brisbane from a sub-tropical region, and Mandurah, Mildura, Gippsland, Renmark, Murray Bridge, and Coorong from a Mediterranean region.** The basemap data were collected from the Australian Bureau of Statistics ('States and Territories—2021 –Shapefile', https://www.abs.gov.au/statistics/standards/australian-statistical-geography-standard-asgs-edition-3/jul2021-jun2026/access-and-downloads/digital-boundary-files).

(Queensland), Mildura and Gippsland (Victoria), and Coorong, Murray Bridge, and Renmark (South Australia) (Fig 1). Each site comprised different study areas, as follows: in Mandurah, the Local Government Areas (LGA) of Rockingham, Mandurah, Murray, and Waroona; in Darwin, the Statistical Areas (SLA) level 2 of Alawa, Anula, Brinkin-Nakara, Coconut Grove, Jingili, Karama, Leanyer, Lyons, Malak, Marrara, Millner, Moil, Nightcliff, Rapid Creek, Tiwi, Wagaman, Wanguri, and Wulagi; in Brisbane, the Brisbane LGA; in Mildura, the SLA level 2 areas of Mildura (North and South), Red Cliffs, Irymple, and Merbein; in Gippsland, the SLA level 2 areas of Wellington, Sale, Longford, Paynesville, and Bairnsdale; in Renmark, the Renmark Paringa LGA; in Murray Bridge, the Murray Bridge LGA; and in Coorong, the Coorong LGA. These LGAs and SLAs are defined by the Australian Bureau of Statistics and make up the catchment areas where mosquito monitoring programs are actively carried out. These areas were selected because they capture the human populations in proximity with the

mosquito populations being monitored. All data used in the following sites and models were collected between 1991–2014 for Mandurah, Darwin and Mildura, 1998–2013 for Brisbane, 1996–2013 for Gippsland, and 1999–2017 for Renmark, Murray Bridge, and Coorong.

Common names, genus and species and reference, hosts include AW, Agile wallaby (*Macropus agilis*); EGK, Eastern grey kangaroo (*Macropus giganteus*); WGK, Western grey kangaroo (*Macropus fuliginosus*); RK, red kangaroo (*Macropus rufus*); BTP, Brush-tailed possum (*Trichosurus vulpecula*); and vector include: AeV, *Aedes vigilax*; CuA, *Culex annulirostris*; AeC, *Aedes camptorhynchus*; and CuG, *Culex globocoxitus*. Host species and density estimates (and confidence intervals) were sourced from the literature as follows: Darwin [28–32]; Brisbane [8,32–34]; Mandurah [8,24,31,32]; Mildura [32,35–38]; Gippsland [32,35–37]; and Renmark, Murray Bridge, and Coorong [8,32,39–41].

## Human incidence and population data

Data for de-identified RRV human notifications per week were provided for each of our study sites by State and Territory from the following sources: in Western Australia, Department of Health and the Western Australian Notifiable Infectious Diseases Database; in Northern Territory, Northern Territory Department of Health and Centre of Disease Control Surveillance Unit; in Queensland, Queensland Department of Health and the Communicable Disease Branch; in Victoria, Victorian Department of Health and the Communicable Disease Epidemiology and Surveillance Branch; and in South Australia, Department of Health South Australia and the Communicable Disease Control Branch. Annual population data were collected from historical and current records from the Australian Bureau of Statistics for each study site [42]. Annual human population data were interpolated, using piecewise linear interpolation, into weekly observations within each year. Using human population data collected from the Australian Bureau of Statistics [32], we calculated RRV notifications per 100,000 individuals per week, which was used to fit the deterministic models.

RRV human notification data were combined with mosquito surveillance data by week and year (vector monitoring data is detailed below). Where mosquito monitoring was unavailable (ranging from 0% to 78% missing) for weeks with RRV notifications, mosquito species densities interpolated using cubic spline interpolation. Our data also retained observations where mosquito monitoring had been conducted, but no RRV notifications were observed.

## Vector monitoring and competence data

Weekly mosquito trapping count data were provided by each respective State and Territory's mosquito surveillance and control programs, including The Arbovirus Surveillance and Research Laboratory for Western Australia, The Centre for Disease Control for the Northern Territory; the Mosquito Management and Asset Services in the Brisbane City Council for Queensland; the Department of Primary Industries for Victoria; and the University of South Australia for South Australia. Two vector species were used in the following models based on their known relative importance in RRV transmission and relative abundance in each site.

There are four major mosquito vector species responsible for the majority of RRV transmission: *Aedes vigilax*, *Culex annulirostris*, *Aedes camptorhynchus*, and *Aedes notoscriptus*. These species also typically make up the greatest proportion of mosquito community abundance in epidemic centres, supporting our preliminary analysis. We restricted our analysis to two species of vector mosquito to prevent overfitting our models, as at each site, the other species of mosquito were at very low densities in comparison (See S1 Appendix for species abundance over time). The relative abundance and dynamics of these major RRV vector species can vary across regions based on climate and habitat. For example, *Ae. vigilax*, *Cu. annulirostris*, and

**Table 2. Invariant population parameters shared across sites of host and vector species recovery rate of infection, general mortality rates, and vertical transmission of RRV in vector populations.**

| Population | Parameter | Rate Definition (frequency) | Value (Range) | Source |
|---|---|---|---|---|
| *Vector* | $\phi_{v_i}$ | Vertical transmission & larval survival (daily) | $8.5 \times 10^{-4}$ | [35] |
| | $d_{v_i}$ | Mortality rate (daily) | 0.10 (0.02–0.33) | [43] |
| *Human* | $\gamma_h$ | Recovery rate (annual) | $\frac{365.25}{4(1-6)}$ | [34,53,54] |
| *Macropods* | $\gamma_k$ | Recovery rate (annual) | $\frac{365.25}{6(2-6)}$ | [53] |
| | $d_k$ | Mortality rate (annual) | $log(1+b(t)_k)$ | A relatively stable population over time |
| *Possums* | $\gamma_p$ | Recovery rate (annual) | $\frac{365.25}{1(1-3)}$ | [55,56] |
| | $d_p$ | Mortality rate (annual) | $log(1+b(t)_p)$ | A relatively stable population over time |

*Ae. camptorhynchus* differ in host feeding preferences and the timing of population peaks. We selected the most relatively abundant species at each site as likely important to local transmission patterns. The selection of only two vector mosquitoes in these models does not reflect that other mosquito species do not contribute to RRV transmission. This choice was top balance model complexity and the low populations of other vector species.

Each site had several mosquito trapping stations used to capture mosquitoes, and we calculated the mean weekly number of mosquito species per trap. The mosquito species used for each site were then selected based on their relative abundance within the mosquito species community. The mosquito species with the greatest maximum abundance was classified as the primary vector and the second most abundant as the secondary vector. The use of the term "secondary vector" does not indicate that this vector is any more of less important for transmission than the primary vectors, it merely represents a second vectors species in the transmission system being investigated. It is assumed that these mosquito species populations do not become extinct but have a constant mortality rate of 10% per day (Table 2) [41,43–49]. The primary and secondary vectors for each site can be found in Table 1.

Our models estimate the maximum and minimum mosquito population density per week, denoted by $v_{1max}$ and $v_{1min}$, respectively, which populations could not exceed or fall below. Furthermore, the relative abundance of the secondary mosquito population, denoted by $v_2$, compared with the primary vector species was also estimated. The parameter $v_{1min}$ was constrained such that it could not exceed 1% of the maximum of primary vector species based on raw mosquito trap data, which is an average across all the traps at each site. Furthermore, secondary mosquito vector populations are scaled to that of the maximum observed primary vector population size in the models presented here. Vertical transmission in vector mosquitoes varies, having different effects on the epidemiology of diseases by location influencing disease persistence, transmission, and magnitude of epidemics [35,50–52]. We consider RRV vertical transmission, denoted as $\phi_{v_i}$ (Table 2), among our vectors of 0.85 per 1,000 mosquitoes using known estimates from *Aedes camptorhynchus* [35] and generalised to the other mosquito species investigated here. Empirical mosquito population data were used to simulate vector abundance through time, allowing for parametrisation of an estimated birth rate for our vector species (Eq 3).

## Reservoir host population and competence data

Host species information used in the following models is detailed in Table 1. Host population data are often limited, giving a single snapshot of a population's abundance with no information regarding temporal change [8]. Therefore, we use a point estimate for animal host abundance when parametrising our models (Table 1). Given the paucity of host population data

available and the relative complexity around determining the potential host species involved in RRV transmission [2–4], we focused on only two host reservoirs in our transmission models. Macropods and possums were selected owing to their known competence and relatively high abundance in the epidemic centres investigated here [8]. Whilst there are other ecologically important hosts that also contribute to RRV transmission (i.e., birds) and add to the ecological complexity in mechanisms driving disease occurrence, we used marsupial hosts owing to their known role in RRV transmission and our ability to reasonably approximate their populations. Where possible, species densities are taken from peer-reviewed literature and converted into a density representing the number of individual species per km$^2$. For species where exact densities were unavailable (such as with possums), we made approximations using methods developed in Koolhof & Carver (2017) based on the number of households within each site to then derive species densities per km$^2$. Host mortality rates vary greatly depending on multiple seasonal, environmental, and climatic differences. Owing to this variability and lack of continuous and reliable mortality estimates, we assume relatively stable host populations through time (Table 2). It is assumed that the recovery rate of an RRV infection to be the average number of days in the year (365.25) divided by the viremic duration in days of host species $i$ (Table 2). The host viremic periods used here are based on empirically founded estimates and do not account for variation in length of viremia, nor do we account for viral titres.

Monthly host breeding estimates (percent of the population breeding) were collected from the literature closest to our study sites in Darwin [31,57], Brisbane [58], with macropod information coming from Victorian estimates [59,60], Mandurah [61–64], and Mildura, Gippsland, Renmark, Murray Bridge, and Coorong [37,59,60,65]. For Darwin, marsupial reproduction is typically seen year-round, reflective of the tropical climate, and thus was not considered to have seasonal breeding (Table 1). Breeding estimates were adjusted by the average time between conception and being 'young at foot' (out of the pouch and exposed to feeding vectors) for macropods, and the time it takes for possums to become back riders; 7–9 and 5 months, respectively. The relative amplitudes of the seasonally forced birth rates are denoted by $\delta_i$ for species $i$ (Eq 3). Moreover, the timing of host reproduction varies latitudinally in Australia, with the reproduction and breeding of hosts in southern latitudes being seasonally driven when compared with hosts further north [59]. These models include the forcing of annual seasonal breeding in our southern latitudes, compared with the potential for continuous and less seasonally driven breeding in northern latitudes.

## Model fitting and parameter estimation

Our models fit human RRV incidence data (see Human incidence and population data below) at a weekly level collected at State and Federal levels as part of routine surveillance of nationally notifiable diseases. Parameters were estimated assuming a beta-binomial distribution allowing for overdispersion, and model section on maximum likelihood estimations. RRV has long been regarded to be considerably underreported/undiagnosed, with wide variation in the severity of symptoms RRV infections cause, asymptomatic infections, and individual behavioural differences in presentations to general practitioners [1,12,66,67]. We account for this variation in reporting by estimating a false-negative reporting rate, more commonly referred to as a notification fraction [68]. Furthermore, as changes in the RRV national case definition in 2006 led to increased false-positive notifications of RRV infections by including non-incident cases, we also estimated a false-positive reporting rate, from 2006 onwards, within our models using a Heaviside step function [69], denoted H(t). Within the beta-binomial, we estimate three additional parameters; a false negative rate denoted by $\beta_r$, a false positive reporting rate of RRV notifications denoted by $\alpha_r$, an overdispersion parameter, denoted by $s$, to account

for variation in reporting dynamics through time and the variation in RRV prevalence (Eq 4). The beta-binomial probability mass function is given in Eq 4, where $f$ is the probability density of the beta-binomial distribution, $\phi$ is the set of model parameters (including $s$), $\varepsilon$ is the number of observed RRV notifications at each reported time point, $n$ is the observed human population size at each time point, $m$ is the number of modelled notifications (which depends on the model parameters $\phi$) at each time point, and $s$ is the overdispersion parameter.

$$m(t) = (1 - \beta_r)I_h + \alpha_r H(t)$$

$$H(t) = \begin{cases} 1 \; if \; t \geq 2006 \\ 0 \; otherwise \end{cases}$$

$$\alpha_r = sm \text{ and } \psi = s(1 - m) \tag{4}$$

$$f(\varepsilon | n, m, s) = \binom{n}{\varepsilon} \frac{B(\varepsilon + \alpha, n - \varepsilon + \psi)}{B(\alpha, \psi)}$$

$$\max_{\phi} L(\phi | \varepsilon) = \max_{\phi} f(\varepsilon | n, m, s)$$

## Model selection

Model combinations were selected based on biologically plausible transmission pathways. Eleven ecological scenarios were fitted to the RRV notification data (Table 3) using combinations of multiple mosquitoes and host species and seasonal vector feeding parameters in deterministic ODE models (Eq 1). Development of these model combinations started with the simplest transmission pathway (e.g., assuming there to be only a single primary vector and human population), then adding increasing complexity which could plausibly contribute to RRV transmission, by considering additional host and vector populations, host recrudescence, and transmission seasonality. Some parameters included in the models could only be included if others were also included. For instance, host recrudescence could only be included if the transmission between vector and host $i$ was also included in the model combination (Table 3). Maximum likelihoods were estimated assuming a beta-binomial distribution using 'lsode', a statistical solver for ordinary differential equations, that utilised a general-purpose optimization method based on Nelder–Mead. Akaike's Information Criterion (AIC) values for each model combination were derived from the maximum likelihood estimates to find and rank the models using delta ($\Delta$) AIC [70]. Parameter estimates were then averaged over all model combinations using model weights derived from the $\Delta$AIC values [71].

To assess the capacity of the mechanistic models to qualitatively capture the dynamics of human RRV incidence, we compared the observed number of outbreaks to the model fitted the number of outbreaks using positive and negative predicted values (PPV and NPV respectively). As our modelling is not for predictive or forecasting purposes, to avoid confusion, we have re-defined these terms here as positive validated outbreak (PVO) and negative validated outbreak (NVO). Here we classified an outbreak of RRV to occur if the number of notifications was above the mean number of RRV notifications per 100,000 plus one standard deviation, calculated over the entire time period for each site [15]. The outbreak analysis using PVO and NVO was intended to provide a qualitative view of whether models could generally

**Table 3. Transmission scenarios model combinations for explaining RRV transmission across epidemic centres in Australia.**

| Scenario | Transmission model description | Parameters estimated |
|---|---|---|
| 1 | Primary vector with humans | $\beta_{vh}$, $v_{1min}$, $v_{1max}$ |
| 2 | Primary vector with humans and kangaroos | $\beta_{vh}$, $\beta_{vk}$, $v_{1min}$, $v_{1max}$ |
| 3 | Primary and secondary vector with humans, and kangaroos | $\beta_{vh}$, $\beta_{vk}$, $v_{1min}$, $v_{1mx}$, $v_2$ |
| 4 | Primary vector with humans, and kangaroos. Seasonality in vector feeding (transmission) | $\beta_{vh}$, $\beta_{vk}$, $v_{1min}$, $v_{1max}$, $\rho_\vartheta$ |
| 5 | Primary vector with humans, kangaroos, and possums. | $\beta_{vh}$, $\beta_{vk}$, $\beta_{vp}$, $v_{1min}$, $v_{1max}$ |
| 6 | Primary and secondary vector with humans, kangaroos, and possums | $\beta_{vh}$, $\beta_{vk}$, $\beta_{vp}$, $v_{1min}$, $v_{1max}$, $v_2$ |
| 7 | Primary vector with humans, kangaroos, and possums. Seasonality in vector feeding (transmission) | $\beta_{vh}$, $\beta_{vk}$, $\beta_{vp}$, $v_{1min}$, $v_{1max}$, $\rho_\vartheta$ |
| 8 | Primary and secondary vector with humans, kangaroos, and possums. Seasonality in vector feeding (transmission) | $\beta_{vh}$, $\beta_{vk}$, $\beta_{vp}$, $v_{1min}$, $v_{1max}$, $v_2$, $\rho_\vartheta$ |
| 9 | Primary vector with humans, kangaroos, and possums. Recrudescence in the kangaroo and possum | $\beta_{vh}$, $\beta_{vk}$, $\beta_{vp}$, $v_{1min}$, $v_{1max}$, $\omega_k$, $\omega_p$ |
| 10 | Primary vector with humans, kangaroos, and possums. Seasonality in vector feeding (transmission). Recrudescence in the kangaroo and possum | $\beta_{vh}$, $\beta_{vk}$, $\beta_{vp}$, $v_{1min}$, $v_{1max}$, $\rho_\vartheta$, $\omega_k$, $\omega_p$ |
| 11 | Primary and secondary vector with humans, kangaroos and possums. Recrudescence in the kangaroo and possum. Seasonality in vector feeding (transmission) | $\beta_{vh}$, $\beta_{vk}$, $\beta_{vp}$, $v_{1min}$, $v_{1max}$, $v_2$, $\rho_\vartheta$, $\omega_k$, $\omega_p$ |

Where, $\beta_{vi}$ the transmission rate from vectors to host species $i$; $\omega_i$ rate of recrudescence in host species $i$; $\epsilon_\mu$ the seasonal phase of transmission; $\sigma_\varphi$ the amplitude of seasonal transmission; $v_{1min}$ minimum mean estimated vector population in proportion to the maximum observed vector population per mosquito monitoring trap per week; $v_{1max}$ estimated maximum mean vector population per mosquito monitoring trap per week; and $v_2$ estimated mean secondary vector population per mosquito monitoring trap per week. Host and vector species $i$ are defined as kangaroos/wallabies, $k$; possums, $p$; and humans, $h$. In each scenario, "X" indicates if a transmission factor was included in the model combination.

capture observed epidemic patterns. It was not used as a quantitative validation metric to assess model fit. Model selection and parameterization were based on maximum likelihood estimation and AIC comparisons of different model combinations. Analyses were performed in R v3.5.3 (R Core Team 2018) using the packages 'lubridate' [72], 'rmutil', 'caret', and 'deSolve' [73] in RStudio (Version 1.2.1335). The ODE solver and R code can be found in the supplementary material (S2 Appendix).

## Results

RRV incidence in Darwin, Brisbane, and Mandurah is near year-round with annual and/or near biennial epidemic seasons (Figs 2 and 3). These areas typically have a build-up of RRV infections over several weeks which characterises epidemic periods, potentially suggesting sustained transmission and spillover from reservoir hosts to human populations. In contrast to these more northern sites, RRV transmission is seasonal in the southern temperate sites of Mildura, Gippsland, Coorong, Murray Bridge, and Renmark, and it is common to have several years between RRV epidemic seasons, with notable periods of no RRV notifications in humans (Figs 2 and 3). Epidemics in these regions occur abruptly with epidemic transmission being short-lived and generally with little reported incidence between epidemics.

We modelled RRV transmission using parameter combinations of vector, host, and transmission (seasonality in host birth and vector feeding, and recrudescence (previously recovered hosts intermittently shedding the virus) in marsupials; see Table A in S3 Appendix for model combinations). Of these models, the most likely (i.e., models with ΔAIC < 4) fitted to RRV

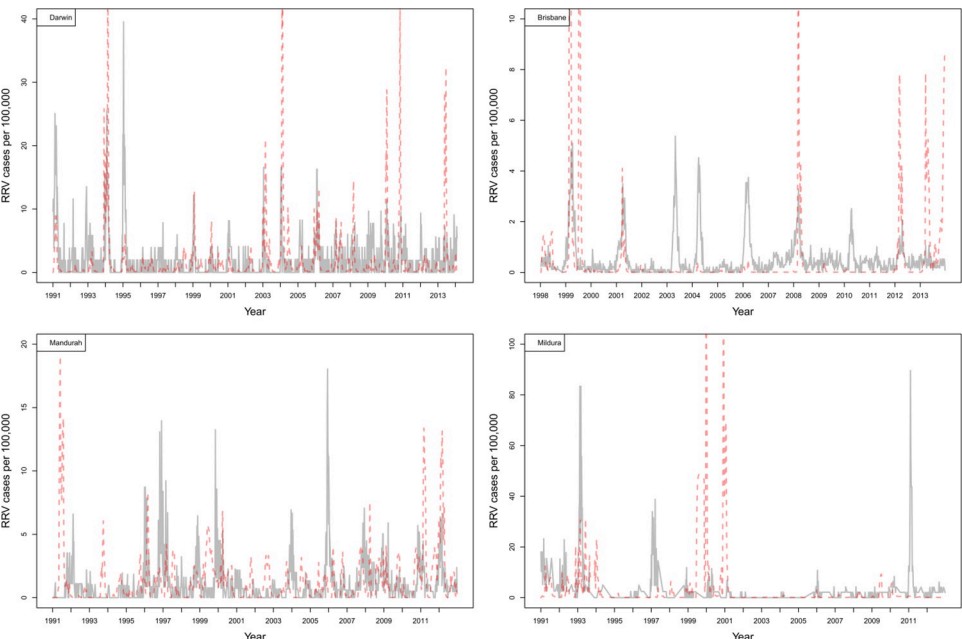

**Fig 2.** Per capita (100,000 people) RRV notifications over time for Darwin, Brisbane, Mandurah, and Mildura showing the recorded data (grey) and weighted model averages from the models (red) (Table 3).

notifications were typical of moderate to high complexity (Table 4). Notably, all most likely models across sites included the primary and secondary vector species and transmission between macropods and possums. Seasonality in vector feeding as a parameter was included among the most likely model for each site, except Mandurah (Table 4). Despite the complexity

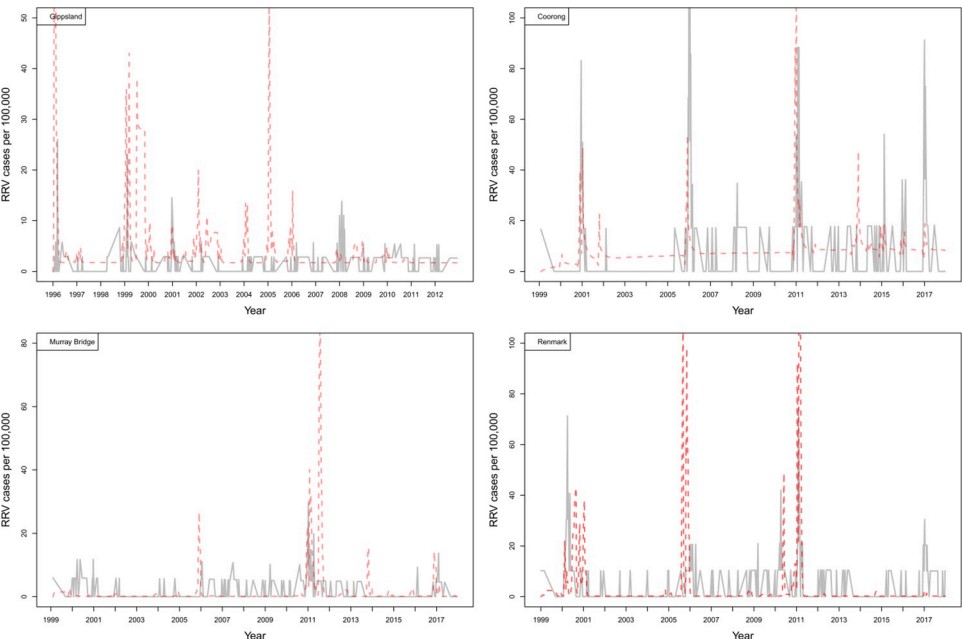

**Fig 3.** Per capita (100,000 people) RRV notifications over time for Coorong, Murray Bridge, and Renmark, showing the recorded data (grey) and weighted model averages from the models (red) (Table 3).

**Table 4. Delta AIC rankings of the importance of vector, host and transmission parameters explored in mechanistic model combinations (Scenarios as outlined in Table 3) fitted to RRV notifications.** Where, $\beta_{vi}$ the transmission rate from vectors to host species $i$; $\rho_\vartheta$ the phase and amplitude of seasonal transmission; $\omega_i$ host recrudescence of host species $i$; $v$ primary vector population; and $v_2$ secondary vector population. Host and vector species $i$ are defined as kangaroos/wallabies, $k$; possums, $p$; and humans, $h$. Best fitting models ($\Delta$AIC < 4) are in bold, with the most likely model fit having a $\Delta$AIC = 0.

| Scenario | Darwin | Brisbane | Mandurah | Mildura | Gippsland | Renmark | Murray Bridge | Coorong |
|---|---|---|---|---|---|---|---|---|
| 1 | 1466.6 | 58.3 | 981.9 | 90.3 | 1276.6 | 194.8 | 270.5 | 352.5 |
| 2 | 1467.9 | 55.3 | 983.9 | 93.3 | 413.2 | 202.0 | 184.7 | 354.4 |
| 3 | 219.5 | 3750.8 | **0.0** | 210.1 | **3.2** | 207.1 | 9.4 | 5.3 |
| 4 | 1471.2 | 57.0 | 986.0 | 299.5 | 1280.5 | 199.2 | 187.3 | 356.4 |
| 5 | 1472.4 | 59.4 | 986.0 | 290.7 | 357.1 | 199.2 | 186.9 | 356.9 |
| 6 | **0.0** | 3801.8 | **1.5** | **0.0** | **0.2** | **0.0** | **0.0** | 359.6 |
| 7 | 1474.2 | 61.5 | 989.0 | 97.3 | 358.7 | 247.6 | 188.8 | 358.4 |
| 8 | **2.0** | **0.0** | **4.0** | **1.7** | **0.0** | **1.9** | **2.5** | 42.6 |
| 9 | 1475.1 | 62.2 | 990.7 | 99.3 | 360.6 | 202.5 | 190.5 | 363.9 |
| 10 | 1479.5 | 5794.0 | 992.4 | 102.2 | 421.0 | 205.9 | 194.0 | 396.8 |
| 11 | 5.3 | **3.7** | 8.8 | 9.0 | 10.2 | 5.4 | 8.8 | **0.0** |

of transmission scenarios examined, there was minimal model uncertainty in parameters included in the most likely models, with two to three most likely models per site ($\Delta$AIC < 4), except Coorong, which had a single best-fit model (Table 4). Recrudescence was included in the most likely models at only two sites Brisbane and Coorong (Table 4). We find that the models were able to generally capture the number RRV notifications leading to observed RRV outbreaks, having high positive predictive values, but were more variable in capturing the non-outbreak dynamics, having low negative predictive values (Table 5). Modelled RRV notifications in Mildura, Gippsland, and Coorong were captured least well, compared to the other sites in non-outbreak periods (Table 5).

## Transmission mechanisms

A valuable contribution of this study is the estimation of host, vector, and seasonal parameters associated with the transmission dynamics of RRV in multiple epidemic areas, allowing for an improved understanding of how mechanisms of transmission systems differ among sites. Despite ecological and environmental differences between the epidemic centres examined, we observed minimal variation in the transmission rate estimates between vector and host populations, particularly between vectors and humans (Table 6).

**Table 5. Capacity of the mechanistic models to qualitatively capture the dynamics of human RRV incidence.** PVO is the positive validated outbreak; and NVO is the negative validated outbreak.

| Site | Mean annual incidence per/100,000 | Incidence pattern | Observed outbreaks | Modelled outbreak | PVO | NVO |
|---|---|---|---|---|---|---|
| **Darwin** | 99.2 | Annual | 132 | 89 | 0.91 | 0.37 |
| **Brisbane** | 26.6 | Biennial | 78 | 67 | 0.91 | 0.21 |
| **Mandurah** | 51.4 | Biennial^ | 82 | 73 | 0.90 | 0.12 |
| **Mildura** | 102.1 | Multi-ennial* | 43 | 27 | 0.92 | 0.07 |
| **Gippsland** | 32.5 | Multi-menial* | 32 | 82 | 0.92 | 0.05 |
| **Renmark** | 108.5 | Multi-ennial* | 35 | 16 | 0.89 | 0.25 |
| **Murray Bridge** | 37.6 | Multi-ennial* | 22 | 13 | 0.94 | 0.39 |
| **Coorong** | 152.1 | Multi-ennial* | 29 | 2 | 0.89 | 0 |

*On average > 4 years between RRV outbreaks, and ^ On average 1–4 years between RRV outbreaks.

**Table 6. Weighted model average parameter estimates of vector, host, and transmission parameters explored in mechanistic model combinations fitted to RRV notifications.** Where, $\beta_{vi}$ the transmission rate from vectors to host species $i$; $\omega_i$ rate of recrudescence in host species $i$; $\epsilon_\mu$ the seasonal phase of transmission; $\sigma_\varphi$ the amplitude of seasonal transmission; $\nu_{1min}$ minimum mean estimated vector population in proportion to the maximum observed vector population per mosquito monitoring trap per week; $\nu_{1max}$ estimated maximum mean vector population per mosquito monitoring trap per week; and $\nu_2$ estimated mean secondary vector population per mosquito monitoring trap per week, $\beta_r$ the false-negative reporting rate (number of unreported infections per one notification), and $\alpha_r$ the false-positive reporting rate (number of false infections reported per one notification). Host and vector species $i$ are defined as kangaroos/wallabies, $k$; possums, $p$; and humans, $h$.

| Parameter | Darwin | Brisbane | Mandurah | Mildura | Gippsland | Renmark | Murray Bridge | Coorong |
|---|---|---|---|---|---|---|---|---|
| $\beta_{vh}$ | $5.24 \times 10^{-4}$ | $8.89 \times 10^{-4}$ | $7.87 \times 10^{-4}$ | $9.65 \times 10^{-4}$ | $3.05 \times 10^{-4}$ | $9.76 \times 10^{-4}$ | $9.88 \times 10^{-4}$ | $9.99 \times 10^{-4}$ |
| $\beta_{vk}$ | $1.24 \times 10^{-4}$ | $1.75 \times 10^{-4}$ | $8.15 \times 10^{-5}$ | $8.08 \times 10^{-6}$ | $4.78 \times 10^{-5}$ | $7.16 \times 10^{-5}$ | $6.96 \times 10^{-7}$ | $9.33 \times 10^{-4}$ |
| $\beta_{vp}$ | $4.40 \times 10^{-5}$ | $1.30 \times 10^{-4}$ | $1.04 \times 10^{-5}$ | $6.98 \times 10^{-6}$ | $4.28 \times 10^{-4}$ | $4.41 \times 10^{-5}$ | $9.75 \times 10^{-6}$ | $8.17 \times 10^{-4}$ |
| $\omega_k$ | $3.68 \times 10^{-2}$ | $3.60 \times 10^{-2}$ | $2.42 \times 10^{-3}$ | $3.58 \times 10^{-3}$ | $9.55 \times 10^{-8}$ | $2.05 \times 10^{-2}$ | $1.15 \times 10^{-3}$ | $9.28 \times 10^{-2}$ |
| $\omega_p$ | $3.52 \times 10^{-2}$ | $2.78 \times 10^{-2}$ | $7.62 \times 10^{-4}$ | $5.67 \times 10^{-3}$ | $2.81 \times 10^{-3}$ | $2.06 \times 10^{-3}$ | $6.36 \times 10^{-3}$ | $1.27 \times 10^{-8}$ |
| $\epsilon_\mu$ | 0 | 0 | $5.60 \times 10^{-3}$ | 0 | 0 | 0 | $8.18 \times 10^{-3}$ | $9.34 \times 10^{-1}$ |
| $\sigma_\varphi$ | $4.44 \times 10^{-01}$ | 1.00 | $9.22 \times 10^{-02}$ | $3.00 \times 10^{-01}$ | $4.77 \times 10^{-01}$ | $3.13 \times 10^{-01}$ | $2.29 \times 10^{-01}$ | $9.34 \times 10^{-01}$ |
| $\nu_{1min}$ | $9.95 \times 10^{-3}$ | $1.00 \times 10^{-2}$ | $1.00 \times 10^{-2}$ | $9.98 \times 10^{-3}$ | $9.94 \times 10^{-3}$ | $9.99 \times 10^{-3}$ | $9.99 \times 10^{-3}$ | $1.91 \times 10^{-3}$ |
| $\nu_{1max}$ | 16 | 70 | 22 | 34 | 7 | 56 | 21 | 2 |
| $\nu_2$ | 5 | 0 | 4 | 2 | 1089 | 2 | 22 | 35901 |
| $\beta_r$ | 0.87 | 0.98 | 0.93 | 0.74 | $1.49 \times 10^{-6}$ | 0.74 | 0.91 | $1.02 \times 10^{-6}$ |
| $\alpha_r$ | $1.19 \times 10^{-11}$ | $3.36 \times 10^{-12}$ | $8.29 \times 10^{-12}$ | $1.07 \times 10^{-9}$ | $2.41 \times 10^{-9}$ | $1.25 \times 10^{-10}$ | $5.04 \times 10^{-12}$ | $1.83 \times 10^{-9}$ |

While seasonality in transmission (representative of vector feeding rate upon hosts) was important in modelling RRV transmission, the phase and amplitude of seasonality and its effect on transmission varied among regions (Table 6). We find seasonal vector feeding commonly amplify RRV transmission across epidemic centres. However, only sites in southern latitudes were found to have regular specific seasons within a year where vector feeding commonly occurs. In Darwin and Brisbane, while there was amplification of RRV transmission resulting from seasonal vector feeding, the timing of this season varied among years, with no common season where vector feeding was more prominent (Table 6).

We found that across sites, for RRV transmission to persist between host and vector populations, the estimated minimum primary vector population generally cannot fall below 0.1% of the maximum observed vector abundance (Table 6). In contrast, the estimated maximum vector population for optimal RRV transmission varied greatly among sites. The primary vector mosquito species chosen for each site were based on the relative species abundance, where greater vector abundance is typically assumed to be a major contributor to viral transmission. However, our analysis showed that despite having a lower relative abundance, in most regions, the secondary vector species still has an important contribution to RRV transmission (Table 6). Despite having a lower relative abundance, in most regions, the secondary vector species still has an important contribution to RRV transmission. This highlights that while the abundance of a vector species is important, the presence of additional secondary vector species, even if less abundant, can still play a key role in transmission dynamics. The consistency of the secondary vector's importance across epidemic regions, which vary in vector species composition, underscores how vector community diversity enables RRV persistence.

Under-reporting (based on false-negative reporting rate) of RRV was estimated to be relatively common across Australia, for every 10 notifications of RRV, there could be between 7.4 to 9.3 additional unreported infections of RRV (i.e., 52–57% of all infections lead to a reported notification), except for in Coorong and Gippsland where the under-reporting rate was less common (Table 6). False-positive notification of RRV was estimated to be very rare in our models and unlikely to influence disease surveillance activities (Table 6).

We also noted that the estimated relative abundance of the secondary vector species and under-reporting rate in Gippsland and Coorong was found to be distinctly different from those seen in other sites, with estimates being biologically unrealistic (Table 6). Moreover, models for Gippsland and Coorong had the greatest uncertainty in capturing non-outbreak dynamics. These factors suggest that the parameterisations may not accurately reflect observed RRV transmission at these sites (Table 5).

## Discussion

Despite being critical to the dynamics and control of mosquito-borne diseases, the most important mechanisms driving disease dynamics in humans are often poorly understood. Here we examined the intrinsic mechanisms in the transmission of Ross River virus (RRV), producing human disease dynamics. We investigated the drivers of transmission between varying epidemic regions in Australia using model selection on our mechanistic ODE models. Our models indicate that the transmission of RRV between vectors and hosts is relatively similar across epidemic regions. The importance of multiple vector and host species was supported across all sites. Seasonal influences on vector biting rates for RRV transmission also appears generally important. Our findings further support RRV incidence as being underreported across Australia, with up to 43% of human infections being estimated as undetected/unreported [67]. Economic impacts of RRV on healthcare and lost productivity are estimated to cost approximately $15 million per annum[1]. However, because of the number of potentially unnotified cases, the true health and economic impact are likely to be more than previously estimated. While underreporting of RRV is expected, we estimate that the false-positive diagnosis of RRV to be infrequent, highlighting the specificity of the current national case definition and laboratory diagnostic procedures.

Zoonotic vector-borne diseases are inherently complex owing to the variety and number of vector and host species that may be involved in pathogen transmission. RRV is no exception, with over 42 known mosquito and 60 vertebrate host species with the potential to contribute to transmission [2,4,6,74]. Intrinsic and extrinsic factors such as vector feeding preferences [4,75,76], relative host competence (i.e., viral titre and viremia), host species community composition [8], host availability to vector mosquitoes, and host abundance, and meteorological and climate conditions [12,13], all contribute to shaping transmission patterns leading to pathogen spillover into humans and mosquito-human-mosquito transmission. This study demonstrates this complexity in transmission dynamics. When deterministically modelling RRV notification, the more complex models that include multiple vectors and host populations and seasonal parameters explain a greater amount of variation in the temporal pattern in RRV transmission than more simplistic transmission scenarios. For example, two vector species, macropod and possum hosts, and seasonality in vector feeding were all supported as important mechanisms for RRV transmission across all eight epidemic sites investigated. In the epidemic centres investigated here, two vector and two reservoir host populations, viral recrudescence in host populations, and seasonality in RRV transmission were assessed in 11 scenario model combinations likely driving transmission. Additional model combinations not considered here may also drive transmission (e.g., other vector or host species). However, the consistency in the best fit models among our sites gives us confidence that these model combinations are a reasonable representation of the transmission dynamics based on the parameters used. This is further supported by minimal variation among our transmission parameter estimates, particularly the vector to human transmission rate ($\beta_{vh}$).

Past investigations of host contributions as RRV reservoirs have generally relied on serological studies detailing relative host competence, which greatly differs among species [1,7,77].

We explored vector-host transmission rates, finding minimal variation in RRV transmission rates between vector and humans ($\beta_{vh}$). Similar patterns in macropod and possum hosts were also observed, with macropod and possum species ($\beta_{vk}$, $\beta_{vp}$) having similar transmission rates to one another among our sites (although more variation than $\beta_{vh}$). This reinforces that, despite the relative competence and abundance of macropods, other species such as possums can contribute to the transmission and amplification of RRV. This supports previous studies, finding that secondary hosts such as possums and host species community compositions may amplify RRV transmission [2,8]. Our analysis showed the secondary vector species contributed to transmission despite having lower relative abundance compared to the primary species. This supports the importance of vector diversity in enabling RRV persistence, rather than abundance of a single dominant species. Moreover, our findings further support urban RRV transmission in built-up areas with few macropods but many possums. Species outside marsupials, such as placental mammals and birds, are increasingly being considered as having the potential to contribute to RRV transmission dynamics[2,4], and further research to explore their roles would be of value.

We tested the recrudescence of RRV as a likely explanation of the mechanisms involved in RRV persistence, particularly during winter periods when vector populations decline and in southern temperate regions where RRV notifications in humans are infrequent, with multi-annual periods between epidemics. In most cases, we find that the viral recrudescence of RRV in hosts was not supported to contribute to RRV transmission. Environmental and seasonal stress is thought to drive immunosuppression allowing for the re-emergence and short-term viremic periods in recovered hosts that may facilitate the reintroduction and circulation of RRV. According to our models, viral recrudescence in hosts likely plays a lesser role in the general transmission of RRV, except for potentially in Coorong and Brisbane.

Generalist feeding mosquitoes display high plasticity in feeding patterns on host species which vary by ecological setting and host community composition and abundance [76,78–80]. Seasonal and environmental factors are associated with vector abundance and are often used in deterministic and predictive modelling for RRV [12,13,81–83]. Our finding on the role of seasonality in vectors transmission, as an indicator of seasonal variation in vector feeding, suggest seasonal amplification commonly occurs across epidemic regions, but that the timing of seasonality in vector feeding is specific to individual regions. Furthermore, for RRV transmission to persist, we found that vector abundance cannot fall below 0.1% of the maximum observed vector abundance. Understanding how small vector populations effectively restrict RRV transmission may prove useful for mosquito control strategies. Because of logistical and cost constraints, mosquito surveillance is often targeted to specific times of the year when disease risk is greatest and for the domestic nuisance caused by mosquito populations. Because of this, vector abundance and its importance in sustaining RRV transmission is relatively unknown for parts of the year when their abundance is lowest. In contrast to common understanding, we also find that the transmission of RRV within our models does not require large population densities of vectors for effective transmission to humans.

Host competence is well documented in serological studies of RRV [55,84]. However, more empirical information on host community structures and host life-history traits associated with transmission are needed to better understand the role that hosts play in RRV transmission. We found transmission rates between vector and host populations were relatively similar across epidemic regions. Where variation between areas was observed, it was largely transmission rates associated with the reservoir hosts and vectors than between vectors and human populations. Here we make assumptions on host densities due to the significant gap in general host abundance data and the lack of temporal surveillance of wildlife host populations. For instance, we assume hosts have relatively stable populations through time, although marsupials

in these regions do undergo population changes [33,39,40,60–62]. Monitoring of host populations could be considered for inclusion in RRV surveillance and monitoring. Notwithstanding these limitations, we bring together the most available realistic combination of host abundance and reproduction likely contributing to transmission. The consistency in the vector to host transmission rates gives confidence they are at least representative of the transmission ecology of RRV.

Our findings suggest seasonal vector feeding influences host-vector transmission, leading to RRV spillover into humans. However, the seasonal phase in which vector feeding contributes to transmission varies across Australia. For instance, the transmission of RRV in Darwin and Brisbane both experience an amplification of RRV from seasonal vector feeding; however, there is no consistent time of year this occurs. We hypothesise, in this instance, areas such as Darwin and to a lesser extent Brisbane, have favourable seasonal and climatic conditions allowing for continuous host breeding, with no distinct breeding season, and with vector populations being present throughout the year with well-defined annual cycles in vector populations. Therefore, RRV transmission can more readily occur year-round. This is in contrast with temperate areas where host and vector populations are heavily driven by environmental patterns spanning several years.

A significant strength of our study is the bringing together of diverse ecological and epidemiological data to describe in depth the mechanistic processes driving RRV transmission. To the best of our knowledge, this is the first study to bring together long-term empirical disease surveillance data to test potential mechanisms driving disease patterns in RRV human incidence. Mechanistic models rely upon understanding several aspects of pathogen, vector, and host interactions to begin determining rates of transmission and the dynamics that underpin these rates. The collection of this heterogeneous data from the literature has limitations [85,86]. For instance, here vertical transmission rates in vector populations are based on laboratory studies conducted on *Aedes camptorhynchus* and are generalised across the vector species used here.

A limitation of our deterministic modelling approach is that measures of uncertainty, such as confidence intervals, are not generated for the parameter estimates. A key outcome this study and our findings is the ability for future research to begin using the parameters estimated here to conduct sensitivity analysis on key parameter important in RRV transmission, such as host and vector populations, to determine the impact varying these populations have on RRV notifications. By conducting this future research, we can begin to assess how many RRV cases are averted by current vector management strategies and evaluate the cost-benefit of varying levels of mitigation. Furthermore, our models do not account for variation in the relative viral titre in hosts. Variation within this parameter may alter transmission. While our models grouped macropods into a single reservoir host category, we incorporated species-level differences in density, viremic period, and birth seasonality based on available empirical evidence. Moreover, we also modelled seasonal macropod births using site-specific data on breeding patterns. These parameterised differences between macropod species and sites capture some of the variability that could drive their transmission roles. Quantifying these variable characteristics between macropod species, and their subsequent impacts on RRV dynamics, represents an important direction for future research. Our current models balance parsimony with incorporating measurable differences between species and sites in density, viremia, and births. But as evidence on other species-specific factors accumulates, models could aim to represent additional sources of variation that may alter transmission. While we attempted to collect the most accurate and precise host population and reproductive estimates specific to each site, these population data represent a static "snapshot". Due to the lack of continuous host population data, our mechanistic models do not adequately capture the fluctuations seen the marsupial

and macropod populations. Here we utilise the most of what host-reservoir population data can be pulled together across our sites. Long term host population surveillance would provide better insight into enzootic transmission, the benefits, logistics and costs associated with regular monitoring would need to be carefully considered. Future research could address this gap by conducting site-specific hosts monitoring programs to inform and improve deterministic and predictive modelling capabilities. Moreover, we further acknowledge that there are other host species that play important roles in RRV transmission. While testing these other hosts was beyond the scope of this study, future research could test other species to better understand their relative role and contribution to RRV circulation. Mechanistic differential equation models allow a large degree of flexibility in parameter estimation and in the fitting of disease distributions despite some limitations. Models used here show that while relying on time-point estimates of host densities, patterns of disease transmission can still be determined. Mosquito-borne diseases also circulate through multiple vectors and hosts. Here we have simplified transmission with two vector and two host species to prevent overfitting models to obtain transmission rate estimates between these populations. This does not mean to say that additional hosts and vector species not included here do not contribute to RRV transmission. With these new estimated vector and host transmission parameters, future studies can begin to expand and assess further aspects to RRV transmission, including additional vector and host species, and how management practices influence disease dynamics in these epidemic centres.

The modelling presented here accounts for the underreporting of RRV infections, which is an important distinction when understanding true ecological processes in disease transmissions. From an ecological perspective, understanding total disease transmission provides insights into the mechanisms and extent of virus spread between vectors and hosts. However, for practical applications like resource allocation, forecasting reported notifications has greater utility, although uncertainty around underreporting makes predicting true case numbers difficult. Our aim was to investigate RRV transmission dynamics, not develop a forecasting tool per se. We provide a retrospective analysis estimating underreporting rates, which has been lacking for RRV and is valuable ecologically. However, a model directly forecasting notifications would better inform public health planning, albeit with uncertainty about true infections. It is possible that future work might look to use a similar mechanistic modelling approach for forecasting (in addition to the statistical approaches already used), but that would likely need to be after more mechanisms have been explored and a mechanistic basis more settled upon. Approaches separating the infection process from reporting could be beneficial in future work to separately estimate underreporting for ecological understanding and predict notifications for mitigation planning. Investigating how variation in vector control and diagnostics has influenced historical reporting would further improve interpretation of notifications.

In this study we brought together the most diverse set of ecological and epidemiological data for RRV using long-term surveillance to describe the most in-depth understanding of mechanisms driving patterns of human incidence across epidemiologically important areas of Australia. We enhance current understandings of the ecology underlying RRV transmission mechanisms which may be used in public health management. Our results highlight the complex nature in RRV transmission dynamics, and the extent of under-reporting/undiagnosed RRV infections across Australia. The information gained here can be used to inform future research to improve understandings of vector control programs beneficial in reducing health and economic impacts. The establishment of empirically founded mechanistic models fit to empirical data is a critical frontier to effectively evaluate factors influencing mosquito-borne disease dynamics in existing and new epidemic areas.

## Supporting information

**S1 Appendix. Vector mosquito population trend.** Mosquito population trends by mosquito species and study site.
(DOCX)

**S2 Appendix. R and solver code.** Code and solver used to run the ordinary differential equations.
(DOCX)

**S3 Appendix. Transmission scenarios.** Transmission scenarios model combinations for explaining RRV transmission across epidemic centres in Australia.
(DOCX)

## Acknowledgments

Mosquito data used this research was supported by Health Departments and Local Councils and the staff within these bodies who have undertaken the mosquito monitoring programs spanning several years. We would like to thank the Departments and staff involved who have contributed to the mosquito surveillance used here, which includes the Arbovirus Surveillance and Research Laboratory at in the School of Pathology and Laboratory Medicine at the University of Western Australia, and the Department of Health, Western Australia; Victorian Department of Health and Human Services; AgriBio Agriculture Victoria; Centre for Disease Control, Northern Territory Department of Health; Brisbane City Council; Department of Environment, Land, Water and Planning, Victoria; Clinical & Health Sciences, University of South Australia. Moreover, RRV host ecology information (such as reproductive timing and population abundance) for Victoria and South Australia was provided by Graeme Maxwell (University of Melbourne) and Allison Mckay.

## Author Contributions

**Conceptualization:** Nicholas Beeton, Scott Carver.

**Data curation:** Iain S. Koolhof, Nicholas Beeton, Katherine Gibney, Peter Neville, Andrew Jardine, Peter Markey, Nina Kurucz, Allan Warchot, Vicki Krause, Michael Onn, Stacey Rowe, Lucinda Franklin, Stephen Fricker, Craig Williams, Scott Carver.

**Formal analysis:** Iain S. Koolhof, Nicholas Beeton.

**Investigation:** Iain S. Koolhof, Silvana Bettiol, Michael Charleston, Simon M. Firestone, Peter Neville, Andrew Jardine, Peter Markey, Nina Kurucz, Allan Warchot, Vicki Krause, Michael Onn, Stacey Rowe, Lucinda Franklin, Stephen Fricker, Craig Williams, Scott Carver.

**Methodology:** Iain S. Koolhof, Nicholas Beeton, Michael Charleston, Simon M. Firestone, Scott Carver.

**Project administration:** Iain S. Koolhof, Silvana Bettiol, Scott Carver.

**Software:** Iain S. Koolhof.

**Supervision:** Silvana Bettiol, Michael Charleston, Simon M. Firestone, Scott Carver.

**Visualization:** Iain S. Koolhof.

**Writing – original draft:** Iain S. Koolhof, Andrew Jardine, Scott Carver.

**Writing – review & editing:** Iain S. Koolhof, Nicholas Beeton, Silvana Bettiol, Michael Charleston, Simon M. Firestone, Katherine Gibney, Peter Neville, Andrew Jardine, Peter Markey, Nina Kurucz, Allan Warchot, Vicki Krause, Michael Onn, Stacey Rowe, Lucinda Franklin, Stephen Fricker, Craig Williams, Scott Carver.

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
