## [Decision Letter · Decision Letter 0]

17 Aug 2023

Dear Mr Koolhof,

Thank you very much for submitting your manuscript "Testing the intrinsic mechanisms driving the dynamics of Ross River Virus across Australia" for consideration at PLOS Pathogens. As with all papers reviewed by the journal, your manuscript was reviewed by members of the editorial board and by several independent reviewers. In light of the reviews (below this email), we would like to invite the resubmission of a significantly-revised version that takes into account the reviewers' comments.

The article has now been reviewed by three experts in the field. Both reviewer 1 and 3 saw merit in the study and have provided suggestion for improvement. Reviewer 2 was less optimistic and raised a number of concerns. Concerns around the paucity of data being fed into the models and the potential impacts of that should be addressed.

We cannot make any decision about publication until we have seen the revised manuscript and your response to the reviewers' comments. Your revised manuscript is also likely to be sent to reviewers for further evaluation.

Sincerely,

Michelle Wille

Guest Editor

PLOS Pathogens

Alexander Gorbalenya

Section Editor

PLOS Pathogens

Kasturi Haldar

Editor-in-Chief

PLOS Pathogens

orcid.org/0000-0001-5065-158X

Michael Malim

Editor-in-Chief

PLOS Pathogens

orcid.org/0000-0002-7699-2064

The article has now been reviewed by three experts in the field. Both reviewer 1 and 3 saw merit in the study and have provided suggestion for improvement. Reviewer 2 was less optimistic and raised a number of concerns. Concerns around the paucity of data being fed into the models and the potential impacts of that should be addressed.

Reviewer's Responses to Questions

**Part I - Summary**

Reviewer #1: I have reviewed he manuscript titled “Testing the intrinsic mechanisms driving the dynamics of Ross River Virus across Australia” submitted to PLOS Pathogens.

The manuscript describes the use of modelling to determine the mechanisms most likely driving activity of mosquito-borne Ross River virus in Australia with an focus on regions where disease is often greatest. Investigations of this nature are particularly useful to local health authorities in developing response strategies to this and other mosquito-borne diseases.

Firstly, it would be a useful addition to the manuscript to add a brief description of the clinical aspects of the disease. While there are many papers on the topic, it would be beneficial here for readers unfamiliar with the disease. I note that there are already papers cited by authors that include clinical aspects. The brief descriptions could be added to second paragraph of Introduction.

Reviewer #2: I very much appreciate the aims of the paper. I also appreciate the approach taken by comparing a set of models of increasing complexity. However, not having models within the set of alternative models that account for interannual variation makes little sense to me within the Australian context. If there is one thing characteristic for candidate RRV vector and notably host species, it is their opportunistic life histories, resulting in (very) large population changes varying in concert with the erratic climatic conditions on the Australian continent. Fluctuations that may supersede any seasonal variations. There is a wealth of literature on (arbo) virus dynamics on the Australian continent that illustrate these multi-year dynamics.

I am also slightly disappointed in the learnings that we can take out of the presented modelling exercise. Authors claim that “The information gained here can be used to improve the management of disease dynamics beneficial in reducing health and economic impacts”, without providing clear avenues on how exactly this research could be used to that effect; this is entirely left to the creativity of the reader.

I concur with the authors that there is a paucity of information on the composition of the RRV reservoir and vector communities and on the role individual members of those communities might have in RRV dynamics. This almost inevitably means that it will be difficult to make predictive models. However, in the absence of accurate data, mathematical models can still be very valuable to identify the most crucial gaps and help prioritise empirical research and monitoring. Unfortunately, I failed to discover such guidance for future research in this manuscript.

Reviewer #3: This manuscript presents results of mechanistic SIR models fitted to Ross River Virus human notification data, with ancillary data from vector surveillance programs and wildlife abundance estimates used to investigate the support for different levels of transmission complexity across several Australian RRV epicentres. The study is nicely written and presented, though I am not an epidemiologist so cannot comment on the validity of the differential equations. However I do have some comments that I feel can improve the clarity and presentation of the work, so I have listed them below.

**Part II – Major Issues: Key Experiments Required for Acceptance**

Reviewer #1: I have no comments on major issues associated with the paper.

Reviewer #2: “The primary aims of this study are to (1) disentangle key mechanisms responsible for driving observed human RRV dynamics; (2) estimate key parameters associated with the transmission of RRV and reporting of RRV notifications; and (3) assess how mechanisms driving RRV dynamics vary, or are similar, among epidemic sites around Australia.” Yet, how can these be investigated when there are such uncertainties regarding hosts and vector species and dynamics? Instead, I suggest focussing on conducting a sensitivity analysis to identify the most crucial model parameters that warrant further empirical research.

Also see Part I

Reviewer #3: (No Response)

**Part III – Minor Issues: Editorial and Data Presentation Modifications**

Reviewer #1: It may be worth noting in the description of study sites within Methods section that the eight locations included in study are all likely to have different communities of mosquitoes driving RRV transmission. As a consequence, climatic drivers of mosquito abundance will also differ. I note this is discussed later in manuscript but a state here would be useful. There is also a need to clearly state that the four mosquito species selected all vary greatly in their population dynamics, habitat associated and, to some degree, their host feeding preferences as currently known. Perhaps a brief statement on these issues could be included in the “vector monitoring and competence data” section of Methods would be useful.

The authors do include a comment about the potential for birds to be involved as reservoir hosts. It would be good to clarify in methods why bird density, or some other measure of bird diversity, was included. I am not proposing that it should have been included in analysis but a clear statement why it wasn’t would be useful.

In interpreting the results, I would be interested in the authors’ analysis of importance of having multiple vectors and differences in their relative abundance at each location. For example, some inland areas would only have two species present while other would have multiple (perhaps all four). Seasonality of species will also vary with species overlapping in temporal abundance while others less so. Does the analysis reveal the key mosquito species at each location?

It would be interesting to see the authors comment on the potential differences should individual macropod species be isolated in analysis. For example, while there is a paucity of information about the specific role of species in RRV transmission, could there be differences related to population dynamics of different macropod species or groups, such as wallabies or kangaroos? There is some discussion of host contributions from the analysis but this point is of interest.

A final minor comment. Authors should check journal requirements for abbreviation of scientific names. Also, a review of format of references cited is required to ensure consistency.

Reviewer #2: The abstract was rather vague on what was researched and what the results are. Could be much more explicit. Examples: “major epidemic centres in Australia” , “We considered the importance of up to two vector species”, “two non-human reservoir host species”, “We found the combination of most important mechanisms to be relatively similar..”.

Table 1: reproduction in Darwin being considered non-seasonal requires better justification.

Line 166: three host species. Should this not be two?

That non-seasonal environmental forcing is lacking from the models (e.g. by assuming population dynamics of both vectors and hosts to be invariable across years) possibly explains why notably “modelled RRV notifications in Mildura, Gippsland, and Coorong were captured least well, compared to the other sites in non-outbreak periods (Table 5).”

Reviewer #3: Line 84: “effects of climate on mosquito species communities”. Can you be more specific about what kinds of effects, i.e. on community composition, relative abundance, seasonal dynamics etc…

Line 121: SIR needs to be defined here

Line 210: I find it strange that the data descriptions are interrupted by the description of the model structure. To me it would make more sense to define the model first, and then describe all data that were used for modelling (i.e. move the sections about study location selection down below the model definition)

Line 225: How many observations of mosquito surveillance data were missing?

Line 357: There is some discrepancy in the original aims of this work and the methods used to validate models. The validation procedure asks whether models fitted to the observed data can reliably reproduce known ‘outbreaks’, but the objective is defined as “obtain the most in-depth mechanistic understanding of transmission dynamics driving patterns of RRV incidence”. I appreciate that a custom model checking function was used, which is a good step to ensuring the model can be useful for a specific purpose, but a better description of the objectives would help to clarify the actual purpose of the model.

On the same topic, wouldn’t it be more useful to compare models based on their out of sample forecast performances? I imagine this is our ultimate reason for employing such a model, to estimate the current state of the system given all available data up to today and then simulate forward trajectories to provide point-based or probabilistic predictions about the likelihood of outbreaks in the coming weeks / months. AIC based on in-sample simulations may not come close to giving a reliable picture of whether these models could be useful for this purpose because estimates of outbreaks were obtained using all data (before the outbreak and after). Could the authors consider implementing a cross-validation exercise to better scrutinise models on their predictive capacities? Or at least provide details describing why they do not think this would be insightful / necessary?

Line 364: It would be much easier to understand the workflow and comment on the analyses if the R code was provided. I realize data cannot be provided, but the code can still be more informative about any decisions that were made than the actual descriptions in the manuscript are. For example, I’m confused about how the PVO and NVO values were calculated. I understand how the observed outbreak values were calculated, but how were the predicted values calculated? Also, did these calculations consider the nonindependence of successive weeks (i.e. if week 30 is an ‘outbreak’ week, wouldn’t week 31 be more likely to also be an outbreak week)?

Figure 2: These plots are hard to decipher, especially since the predictions from the models seem to be cut off by the upper y-axis limit in several plots.

Table 6: It would be useful to report uncertainties of these parameters so any future studies that plan to use them can include more realistic assumptions

Line 466: The findings of high probable underreporting is an important one that also raises questions of how best to validate such models. Do we want models that give an estimate of the true number of disease occurrences, even though they may be different to what is actually observed, or do we want models that can forecast number of notifications to help with resource planning? These are challenging questions but they deserve some attention

PLOS authors have the option to publish the peer review history of their article (what does this mean?). If published, this will include your full peer review and any attached files.

Reviewer #1: No

Reviewer #2: No

Reviewer #3: No
---

## [Decision Letter · Decision Letter 1]

4 Jan 2024

Dear Mr Koolhof,

We are pleased to inform you that your manuscript 'Testing the intrinsic mechanisms driving the dynamics of Ross River Virus across Australia' has been provisionally accepted for publication in PLOS Pathogens.

Best regards,

Alexander E. Gorbalenya

Section Editor

PLOS Pathogens

Kasturi Haldar

Editor-in-Chief

PLOS Pathogens

orcid.org/0000-0001-5065-158X

Michael Malim

Editor-in-Chief

PLOS Pathogens

orcid.org/0000-0002-7699-2064

Editor Comments:

At the proofs stage please correct the wording in Abstract (line 38) and elsewhere that misrepresents RRV as the disease (see https://doi.org/10.1371/journal.pbio.3002130 for discussion of this type of misrepresentation).

Reviewer Comments (if any, and for reference):

Reviewer's Responses to Questions

**Part I - Summary**

Reviewer #1: I have reviewed the revised manuscript and the responses submitted by authors to both reviewer's comments and I consider the manuscript to now be improved. The authors have thoroughly responded to the reviewers comments and I feel have done so satisfactorily and within the scope of the originally conceived research. I feel that this paper is of high quality and can make a substantial contribution to both operational and future research into mosquito-borne disease in Australia. As noted in reviewers' comments, it is notoriously difficult to develop highly accurate models of mosquito-borne disease but there is also great utility in such endeavors as they can be information and identify areas of future attention.

**Part II – Major Issues: Key Experiments Required for Acceptance**

Reviewer #1: No remaining major issues of concern. As stated above, I note authors have addresses concerns raised by reviewers.

**Part III – Minor Issues: Editorial and Data Presentation Modifications**

Reviewer #1: No additional minor issues have been identified.

PLOS authors have the option to publish the peer review history of their article (what does this mean?). If published, this will include your full peer review and any attached files.

Reviewer #1: No

---

## [Editor Report · Acceptance letter]

25 Jan 2024

Dear Mr Koolhof,

We are delighted to inform you that your manuscript, "Testing the intrinsic mechanisms driving the dynamics of Ross River Virus across Australia," has been formally accepted for publication in PLOS Pathogens.

Best regards,

Michael Malim

Editor-in-Chief

PLOS Pathogens

orcid.org/0000-0002-7699-2064